Enhancement of conformational B-cell epitope prediction using CluSMOTE

Solihah Binti binti.solihah@mail.ugm.ac.id 1 2
Azhari Azhari 1
Musdholifah Aina 1
1 Department of Computer Science and Electronics, Faculty of Mathematics and Natural Sciences, Universitas Gadjah Mada , Yogyakarta , Indonesia
2 Department of Informatics Engineering, Universitas Trisakti , Grogol , Jakarta Barat , Indonesia
Ventura Sebastian
Electronic publication date: 2020 Jun 1
Publication date: 2020
Volume: 6
Electronic Location ID: e275
Received 2019 May 12; Accepted 2020 Apr 15
Copyright: ©2020 Solihah et al.
Copyright year: 2020
Copyright holder: Solihah et al.
License: This is an open access article distributed under the terms of the Creative Commons Attribution License, which permits unrestricted use, distribution, reproduction and adaptation in any medium and for any purpose provided that it is properly attributed. For attribution, the original author(s), title, publication source (PeerJ Computer Science) and either DOI or URL of the article must be cited.
License URL: https://creativecommons.org/licenses/by/4.0/

Keywords: Cluster-based undersampling, SMOTE, Class imbalance, Hybrid sampling, Hierarchical DBSCAN, Vaccine design

Funding: Universitas Trisakti (Doctoral scholarship) This work was supported by Universitas Trisakti (Doctoral scholarship). The funders had no role in study design, data collection and analysis, decision to publish, or preparation of the manuscript.

==============================
Background

A conformational B-cell epitope is one of the main components of vaccine design. It contains separate segments in its sequence, which are spatially close in the antigen chain. The availability of Ag-Ab complex data on the Protein Data Bank allows for the development predictive methods. Several epitope prediction models also have been developed, including learning-based methods. However, the performance of the model is still not optimum. The main problem in learning-based prediction models is class imbalance.

Methods

This study proposes CluSMOTE, which is a combination of a cluster-based undersampling method and Synthetic Minority Oversampling Technique. The approach is used to generate other sample data to ensure that the dataset of the conformational epitope is balanced. The Hierarchical DBSCAN algorithm is performed to identify the cluster in the majority class. Some of the randomly selected data is taken from each cluster, considering the oversampling degree, and combined with the minority class data. The balance data is utilized as the training dataset to develop a conformational epitope prediction. Furthermore, two binary classification methods, Support Vector Machine and Decision Tree, are separately used to develop model prediction and to evaluate the performance of CluSMOTE in predicting conformational B-cell epitope. The experiment is focused on determining the best parameter for optimal CluSMOTE. Two independent datasets are used to compare the proposed prediction model with state of the art methods. The first and the second datasets represent the general protein and the glycoprotein antigens respectively.

Result

The experimental result shows that CluSMOTE Decision Tree outperformed the Support Vector Machine in terms of AUC and Gmean as performance measurements. The mean AUC of CluSMOTE Decision Tree in the Kringelum and the SEPPA 3 test sets are 0.83 and 0.766, respectively. This shows that CluSMOTE Decision Tree is better than other methods in the general protein antigen, though comparable with SEPPA 3 in the glycoprotein antigen.

Introduction

A B-cell epitope is among the main components of peptide-based vaccines (Andersen, Nielsen & Lund, 2006; Zhang et al., 2011; Ren et al., 2015). It can be utilized in immunodetection or immunotherapy to induce an immune response (Rubinstein et al., 2008). Many B-cell epitopes are conformational and originate from separate segments of an antigen sequence, forming a spatial neighborhood in the antigen-antibody (Ag–Ab) complex. Identifying epitopes through experiments is tedious and expensive work, and therefore, there is a high risk of failure. Current progress in bioinformatics makes it possible to create vaccine designs through 3D visualization of protein antigen. Many characteristics, including composition, cooperativeness, hydrophobicity, and secondary structure, are considered in identifying potential substances for an epitope (Kringelum et al., 2013). Since no dominant characteristic helps experts to easily distinguish epitopes from other parts of the antigen, the risk of failure is quite high.

The availability of the 3D structure of the Ag–Ab complex in the public domain and computational resources eases the development of predictive models using various methods, including the structure and sequence-based approaches. However, the conformational epitope prediction is still challenging. The structure-based approach can be divided into three, including dominant-characteristic-based, graph-based, and learning-based categories.

There are several characteristic-based approaches, including (1) CEP, which uses solvent-accessibility properties, (2) Discotope using both solvent-accessibility-based properties and epitope log odds ratio of amino acid, (3) PEPITO that adds half-sphere exposure (HSE) to log odds ratio of amino acid in Discotope and (4) Discotope 2.0, which is an improved version of Discotope. It defines the log odd ratios in spatial contexts and adds half-sphere exposure (HSE) as a feature, and (5) SEPPA, which utilizes exposed and adjacent residual characteristics to form a triangle unit patch (Kulkarni-kale, Bhosle & Kolaskar, 2005; Andersen, Nielsen & Lund, 2006; Kringelum et al., 2012; Sun et al., 2009). The dominant-characteristic-based approach is limited by the number of features and the linear relationships between them.

The graph-based method is yet another critical method, although only two from the same study were found during the literature review. Zhao et al. (2012) developed a subgraph that could represent the planar nature of the epitope. Although the model is designed to identify a single epitope, it can also detect multiples. Zhao et al. (2014) used features extracted from both antigens and the Ag–Ab interaction, which is expressed by a coupling graph and later transformed into a general graph.

The learning-based approach utilizes machine-learning to work with a large number of features. It also uses nonlinear relationships between features to optimize model performance. Rubinstein, Mayrose & Pupko (2009) used two Naïve Bayesian classifiers to develop structure-based and sequence-based approaches. SEPPA 2.0 combines amino acid index (AAindex) characteristics in the SEPPA algorithm in the calculation of cluster coefficients (Qi et al., 2014; Kawashima et al., 2008). Aaindex in SEPPA 2.0 is consolidated via Artificial Neural Networks (ANN). However, SEPPA 3.0 adds the glycosylation triangles and glycosylation-related AAindex to SEPPA 2.0 (Zhou et al., 2019). Glycosylation-related AAindex is consolidated to SEPPA 3.0 via ANN. Several researchers utilized the advantages of random forest (Dalkas & Rooman, 2017; Jespersen et al., 2017; Ren et al., 2014; Zhang et al., 2011). The main challenge in developing a conformational B-cell epitope prediction model is the class imbalance problem. This is a condition where the sample of the target or epitope class is less than that of the nontarget or the non-epitope classes.

Several methods have been proposed to handle the class imbalance problem. However, studies that focus on handling this issue in epitope prediction models are still limited. Ren et al. (2014) and Zhang et al. (2011) used simple random undersampling to handle the class imbalance problem. Dalkas & Rooman (2017) used the Support Vector Machine (SVM) Synthetic Minority Over-sampling Technique (SMOTE) method, which is a variant of SMOTE. Another common approach used is weighted SVM, which is included in the cost-sensitive algorithm level category (Ren et al., 2015). Additionally, Zhang et al. (2014) used a cost-sensitive ensemble approach and proved that the method was superior to Easy Ensemble, Balance Cascade and SMOTEBoost (Liu, Wu & Zhou, 2009; Chawla et al., 2003).

Currently, several studies focus on class imbalance using various approaches that are mainly divided into four, including data and algorithm levels, cost-sensitive, and ensemble (Galar et al., 2012). In the data level approach, the resampling method is used to ensure a balanced distribution of data (Gary, 2012). The approaches under this category include undersampling, oversampling and a combination of both (Drummond & Holte, 2003; Estabrooks, Jo & Japcowick, 2004; Chawla et al., 2002; Chawla et al., 2008). In the algorithm level, the minority class is specifically considered. Most algorithms are equipped with a search system to identify rare patterns (Gary, 2012). The learning process of classifiers usually ignores the minority class. Specific recognition algorithms are used to detect rare patterns, providing different misclassification weights between minority and majority classes or different weights (Elkan, 2001; Batuwita & Palade, 2012; Japkowicz, Myers & Gluck, 1995; Raskutti & Kowalczyk, 2003). In general, adding cost to an instance is categorized as cost-sensitive in the data level (Galar et al., 2012). The approach is also applied in the ensemble method (Blaszczynski & Stefanowski, 2014). However, the determination of the weight is carried out through trial and error.

The most common ensemble methods used to handle the class imbalance problem include bagging and boosting. In bagging, a balanced class sample is generated using the bootstrapping mechanism. The sampling methods used in this case include random undersampling and oversampling, as well as SMOTE (Blaszczynski & Stefanowski, 2014; Galar et al., 2012). In boosting, samples are selected iteratively and their weight calculated based on the misclassification costs. Many boosting variations have been proposed, though the most influential is the AdaBoost (Freund & Schapire, 1996).

Random oversampling and undersampling are the simplest sampling methods used in balancing data distribution. Handling class imbalance in the preprocessing data is versatile since it does not depend on the classifier used. Similarly, the random oversampling method is versatile because it does not rely on the classifier used. However, its main drawback is overfitting because new sample data are not added. The SMOTE technique avoids overfitting by interpolating adjacent members of the minority class to create new sample data (Chawla et al., 2002). Furthermore, oversampling that considers certain conditions, such as the density distribution and the position of the sample point to the majority class, improves the performance of the classifier (He & Garcia, 2009; Han et al., 2005). Random undersampling is a concern in the sense that loss of information from the dataset might occur. This is because of pruning may considerably affect and reduce its performance. To reduce the loss of information, several cluster-based methods have been used in resampling (Yen & Lee, 2009; Das, Krishnan & Cook, 2013; Sowah et al., 2016; Tsai et al., 2019).

Cluster-based undersampling can be conducted by omitting class labels (Yen & Lee, 2009; Das, Krishnan & Cook, 2013). Alternatively, it can be performed only on the negative class (Sowah et al., 2016; Lin et al., 2017; Tsai et al., 2019). Das, Krishnan & Cook (2013) discarded the negative class data that overlap the positive in a specific cluster based on the degree of overlapping. According to Yen & Lee (2009), the samples from the negative class are proportional to the ones in the positive class in a particular cluster. Also, Sowah et al. (2016) randomly selected several sample data from each cluster. In Tsai et al. (2019), the cluster members were selected using optimization algorithms. Clustering samples of the negative to positive class may lead to a suboptimal cluster number of data in the negative class (Lin et al., 2017).

In this research, the cluster-based undersampling method is combined with SMOTE to obtain a balanced dataset. The parameter r is defined to determine the proportion of the majority class data sampled and compared with the minority. A classifier model is built with the decision tree (DT) and SVM algorithms to assess the performance of the proposed method.

Material and Methods

Dataset

This research uses Rubinstein’s dataset as training (Rubinstein, Mayrose & Pupko, 2009). The formation criteria of the training dataset are explained by Rubinstein et al. (2008). The study shows the following, (1) The Ag–Ab complex structure should contain antibodies with both heavy and light chains, (2) Contact between antigens and antibodies must occur in the complementarity-determining regions, (3) The amount of antigen residues binds to antibodies is large, and (4) The complex used cannot be similar to other complexes, as stated in the Structural Classification of Proteins criteria (Murzin et al., 1995). The training dataset consists of 76 antigen chains derived from 62 3D structure Ag–Ab complexes. The chain list is shown in Table S1. The complexes are downloaded from the Protein Data Bank (PDB) (Berman et al., 2000).

Two independent test sets are used, including Kringelum and SEPPA 3.0 (Kringelum et al., 2012; Chou et al., 2019). Kringelum’s test set consists of 39 antigen chains. Data were filtered from 52 antigen chains and thirteen antigens were excluded from the list because they were used as training data with the compared method. The data released include 1AFV, 1BGX, 1RVF, 2XTJ, 3FMG, 3G6J, 3GRW, 3H42, 3MJ9, 3RHW, 3RI5, 3RIA, and 3RIF. The details of Zhang’s test set are presented in Table S2A. The test set represents protein antigen in the general category. It is used to compare the CluSMOTE DT with the Discotope 1.2, Ellipro, Epitopia, EPCES, PEPITO and Discotope 2 (Andersen, Nielsen & Lund, 2006; Ponomarenko et al., 2008; Rubinstein, Mayrose & Pupko, 2009; Liang et al., 2009; Sweredoski & Baldi, 2008; Kringelum et al., 2012). The SEPPA 3.0 test set is a glycoprotein category test set. This dataset consists of 98 antigen chains and eight were excluded because they were multiple epitopes, including 5KEM A1, 5KEM A2, 5T3X G1, 5T3X G2, 5TLJ X1, 5TLJ X2, 5TLK X1, and 5TLK X2. The test set was used to compare the CluSMOTE DT with the SEPPA 2.0 SEPPA 3.0, PEPITO, Epitopia, Discotope 2, CBTOPE and BepiPred 2.0 methods (Qi et al., 2014; Ansari & Raghava, 2010; Jespersen et al., 2017). The antigen list for the test set is presented in Table S2B.

Conformational B-cell epitope prediction method

Conformational epitopes are residues of exposed antigens that are spatially close, though they form separate segments when viewed from sequences (Andersen, Nielsen & Lund, 2006). To build a conformational epitope prediction model, the steps needed, as shown in Fig. 1 are include (1) preparing the dataset, (2) balancing the dataset, and (3) creating a classification model for the prediction of residual potential epitopes. The preparation step aims to build the training and testing datasets. The number of exposed residues considered as epitopes is less than the exposed residues that are not-epitopes. Balancing the dataset is meant to overcome the class imbalance found in Step 1, while the classification model categorizes residues as members of the epitope or non-epitope class.

Data preprocessing

The creation of feature vectors and epitope annotations for the training and testing data is conducted on surface residues only. Relatively accessible surface area (RSA) is used as a parameter to distinguish surface and buried residues. Different values were used as limits, including the 0.05, 0.01, 0.1, and 0.25 thresholds (Rubinstein, Mayrose & Pupko, 2009; Zhang et al., 2011; Kringelum et al., 2012; Ren et al., 2014; Dalkas & Rooman, 2017). This variation affects the imbalance ratio between the data epitope and non-epitope classes. Although the standard burial and non-burial threshold are 0.05, the value of 0.01 is used as the limit. This is because of the larger the surface exposure threshold, the smaller the predictive performance (Basu, Bhattacharyya & Banerjee, 2011; Kringelum et al., 2012). Choosing 0.01 as the limit is relevant to the finding of Zheng et al. (2015), where all RSA values of epitopes are positive, though slightly larger than zero.

Figure 1 Development stage of conformational B-cell epitopes prediction.

.

The feature vectors used include accessible surface area (ASA), RSA, depth index (DI), protrusion index (PI), contact number (CN), HSE, quadrant sphere exposure (QSE), AAindex, B factor, and log odds ratio, as shown in Table 1.

ASA and RSA are the key features in determining if a residue is likely to bind to other molecules for accessibility reasons. Although several programs can be used to calculate ASA, the most commonly used include NACCESS and DSSP (Hubbard & Thornton, 1993; Kabsch & Sander, 1983). DSSP only calculates the total ASA per residue, while NACCESS computes the backbone, side chain, polar, and nonpolar ASA. However, NACCESS can only count one molecular structure at a time. These users need to create additional scripts to count several molecular structures at a time (Mihel et al., 2008). This study used the PSAIA application was used (Mihel et al., 2008). The PSAIA is not only limited to counting one molecular structure but can be used to calculate other features, including RSA, PI, and DI. No significant difference is observed between the ASA calculation results using NACCESS and PSAIA. The ASA attribute values used include the backbone, side chain, polar (including oxygen, nitrogen, and phosphorus), and nonpolar atoms (carbon atoms).

Table 1 Features for antigenic determinant and the methods to compute.

Category	Feature	Data Source/Method	
Structural	ASA	PSAIA Mihel et al. (2008)	
	RSA	PSAIA Mihel et al. (2008)	
	Protrusion Index	PSAIA Mihel et al. (2008)	
	CN	Nishikawa & Ooi (1980)	
	HSE	Hamelryck (2005)	
	QSE	Li et al. (2011)	
Physicochemical	AAIndex	Kawashima et al. (2008)	
	B factor	Ren et al. (2014) and Ren et al. (2015)	
Statistic	Log-odd ratio	Andersen, Nielsen & Lund (2006)	
Notes.

ASA solvent-accessible surface area

RSA Relative solvent-accesible Surface Area

CN Contact Number

HSE Half-Sphere Exposure

QSE Quadrant Sphere Exposure

AAIndex Amino Acid Index

RSA is the result of the ASA value with the maximum figure calculated based on the GXG tripeptide theory, where G is glycine and X is the residual sought (Lee & Richards, 1971). The maximum value of ASA is obtained from Tien et al. (2013), which is an improvement of Rost & Sender (1994) and Millerl et al. (1987). There was no difference in the list of datasets obtained using the three methods, as presented in Appendix 1. The RSA attribute values used include the total RSA of all atoms, backbone atoms, side-chain atoms, polar atoms (including oxygen, nitrogen, and phosphorus), and nonpolar atoms (carbon atoms).

DI: The DI of the i th atom refers to its minimum distance to the exposed atoms. The DI attribute values used include the average and standard deviation of all atoms, average of side-chain atoms, maximum, and minimum.

PI: The PI is the ratio of the space of a sphere with a radius of 10 A centered on C α divided by the area occupied by the heavy atoms constituting proteins (Pintar, Carugo & Pongor, 2002). In this study, PI was calculated using the PSAIA software (Mihel et al., 2008). The PI attribute values used include the average and standard deviation of all atoms, average of side-chain atoms, maximum, and minimum.

CN, HSE, QSE: The CN is the total number of C α adjacent to the residue measured under the microsphere environment. It is limited by a ball with radius r centered on C α (Nishikawa & Ooi, 1980). In HSE, the number of C α was distributed in two areas, including the upper hemisphere and the lower hemisphere balls (Hamelryck, 2005). In QSE, the number of C α was specifically distributed in eight regions in the microsphere environment (Li et al., 2011).

AAindex: The AAindex consists of 544 indices representing the amino acid physicochemical and biochemical properties (Kawashima et al., 2008). The AAindex value of each residue was extracted from component I of the HDRATJCI constituents in the aaindex1.txt file. The detail of component I of aaindex file is attached as a Table S3.

B factor: The B factor indicates the flexibility of an atom or a residue. An exposed residue has a larger B factor than a latent residue. The B factor for each atom is derived from the PDB data. The attribute values used include the B factor of C α and the average of all atoms or residues (Ren et al., 2014).

Log odds ratio: This feature is extracted based on the primary protein structure and calculated based on Andersen, Nielsen & Lund (2006). A sliding window of size 9 residues was run in each sequence of antigens in the dataset to form overlapping segments that can be used in the calculation of the appearance of the individual residues. Each segment was grouped as an epitope or non-epitope depending on its center. The log odds ratio was calculated at the fifth position residue based on Nielsen, Lundegaard & Worning (2004). In this study, a segment would be included in the calculation in case the fifth position residue is exposed.

Epitope annotation on the antigen residue is carried out by analyzing the interaction in the PSAIA software (Mihel et al., 2008) using contact criterion, threshold, and Van der Waals radii. The maximum distance of 4 was derived from the chotia.radii file. The ASA change parameters include the Delta ASA, Z_Slice Size, and Probe Radius with values 1.0, 0.25, and 1.4, respectively. The interaction analyzer output is a list of all adjacent residual pairs within the allowable distance range. A procedure for selecting antigen residues that bind to antibodies is created to obtain a list of epitopes.

Handling class imbalance with CluSMOTE

Resampling with undersampling and oversampling has advantages and disadvantages. Therefore, cluster-based sampling was conducted to minimize the loss of information caused by the pruning effect of undersampling. Oversampling with SMOTE has often proven to be reliable. Merging the two increases classifier performance. A parameter stating the degree of oversampling is used to identify the optimal combination.

This study proposed CluSMOTE, a cluster-based undersampling mechanism combined with SMOTE as shown in Fig. 2. Negative class data are clustered using the Hierarchical Density-Based Spatial Clustering of Applications with Noise (HDBSCAN) algorithm. This is meant to identify the optimal clusters based on stability (Campello, Moulavi & Sander, 2013). The number of clusters is less than the positive class data. This means each cluster contains several data. The simplest sampling mechanism is random selection. To select data, the cluster size and degree of oversampling should be considered.

Figure 2 CluSMOTE sampling.

The proposed CluSMOTE method uses the following steps,

1. Separate the positive and the negative class data.

2. Cluster the negative class ( −) using the HDBScan algorithm.

3. Take a certain number of data items from each cluster. Consider the ratio of the number of clusters to the overall members of the negative class. The samples from the Ci cluster is defined in (1), according to Sowah et al. (2016). where MI is the number of minority class samples, MA is the total number of majority classes, M_ci is the number of Ci cluster members, and r is the negative class dataset ratio from the cluster. In case r = 2, the number of negative class datasets to be formed is twice the positive class datasets. The samples are taken from each cluster randomly. (1) Size_Ci=r×MI×M_ci∕MA

4. Combine positive classes with all datasets taken in Step 3.

5. Carry out SMOTE on the results obtained in Step 4.

Program implementation was conducted in the Java programming environment with NetBeans IDE 8.2. A new class for implementing the CluSMOTE method was implemented in the Java Language Programming, supported by the JSAT statistics library version 0.09 (Raff, 2017).

Classification algorithm

Two classification algorithms, SVM and DT, were used to evaluate the performance of CluSMOTE. Generally, SVM is a popular learning algorithm used in previous studies of conformational epitope prediction. DT is often used to handle the class imbalance problem and classified as one of the top 10 data mining algorithms (Galar et al., 2012).

This study uses the JSAT (Raff, 2017) software package, utilizing the Pegasos SVM with a mini-batch linear kernel (Shalev-shwartz, Singer & Srebro, 2007). Pegasos SVM works fast since the primal update process is carried out directly, and no support vector is stored. The default values used for the epoch, regularization, and batch size parameters include 5, 1e−4, and 1, respectively. The decision tree is formed by nodes that are built on the principle of decision stump (Iba & Langley, 1992). Also, the study used a bottom-up pessimistic pruning with error-based pruning from the C4.5 algorithm (Quinland, 1993). The proportion of the data set used for pruning is 0.1.

Performance measurement of the conformational epitope prediction model

A dataset used for conformational epitope prediction contains the class imbalance problem. The area used is mainly under the ROC curve (AUC) as a performance parameter. In class imbalance, the AUC is a better measure than accuracy, which is biased to the majority class. Another performance parameter used is F-measure, as expressed in Eq. (2): (2) Fm=2∗PPV∗SEPPP+SE=2∗TP∕2∗TP+FN+FP

where PPV = TP/(TP + FP) and SE denote sensitivity or TPR. The F-measure is not affected by imbalance conditions provided the training data used are balanced (Batuwita & Palade, 2009). Other metrics that can be used to assess performance include Gmean and adjusted Gmean (AGm). The Gmean is expressed in Eq. (3) below, (3) Gmean=SP∗SE

where SP denotes specificity/TPR and SE denotes sensitivity/FPR. AGm is expressed in Eq. (4): (4) AGm=Gm+SP∗N∕1+NnifSE>00ifSE=0

where Gm is Gmean, SP specificity, SE sensitivity, and Nn the proportion of negative samples in the dataset. AGm is suitable for application in case an increase in TPR is achieved with minimal reduction in TNR. Generally, this criterion is suitable for bioinformatics cases, where errors in the identification of negative classes are unexpected (Batuwita & Palade, 2009). In the case of epitope prediction, the false negative is not expected to be high. The selection of the wrong residue leads to the failure of the subsequent process.

Results and Discussions

The complex-based leave-one-out cross-validation method is used to test the reliability of the classifier model. Each training set is built from the n − 1 complexes and tested with a test set from the n-th complex. Model performance was measured using seven parameters, including TPR, TNR, precision, AUC, Gmean, AGm, and F-measure.

Effect of the selection of the r value on model performance

In the original dataset, the ratio of imbalance between negative and positive classes is 10:1. To assess the effectiveness of sampling, this study utilized several r values derived using the ratio of negative to positive class data. The value r = 1 indicates that only the clustering and undersampling steps are applied. The value r = 2 indicates that the number of negative class datasets is twice the number of positive ones. The test results obtained without a balancing mechanism show the effectiveness of the proposed resampling method. The results of the assessment of the performance of the classification model expressed by the TPR, TNR, precision, AUC, Gmean, AGm, and F-measure parameters are shown in Table 2.

The results of internal model validation on several variations of the r-value are also shown in Table 2. Where the r-value varies from r = 1 to r = 5, both in CluSMOTE DT and CluSMOTE SVM, the TPR and the FPR value tends to decrease with the increase in the r-value. The larger the degree of oversampling, the smaller the TPR. The TNR value, as well as precision, also tends to increase with the increase in the r-value. The increase in TNR values means more negative classes are recognized. This can also be interpreted as TNR value increases means less information loss of the negative class. These two conditions indicate a trade-off between the degrees of oversampling and undersampling. Oversampling without undersampling yields TNR and precision values greater than undersampling without oversampling. Similarly, undersampling without oversampling yields TPR and FPR values greater than oversampling without undersampling. This finding indicates the undersampling mechanism is more effective in increasing positive class recognition than the oversampling, which is consistent with previous studies. Also, the resampling mechanism increases the TPR and FPR values compared to no resampling. However, the overall performance improvement indicated by the AUC, Gmean, AGm, and F-measure is not significant.

Table 2 Performance of classification model with variations in the r-value.

No	Resampling method	r	classifier	TPR (recall)	TNR	Precision (PPV)	FPR	AUC	Gmean	Adjusted Gmean	Fmeasure	
1	Cluster-based only	1	DT	0.855a	0.769	0.454	0.231a	0.812	0.806	0.791	0.581	
2	CluSMOTE	2	DT	0.797	0.834	0.526	0.163	0.815a	0.811a	0.823	0.622	
3	CluSMOTE	3	DT	0.764	0.862	0.558	0.138	0.813	0.807	0.833	0.634	
4	CluSMOTE	4	DT	0.730	0.881	0.575	0.119	0.806	0.796	0.835	0.631	
5	CluSMOTE	5	DT	0.724	0.880	0.591	0.120	0.802	0.794	0.834	0.641	
6	SMOTE only	–	DT	0.644	0.939a	0.732a	0.061	0.791	0.771	0.848a	0.675a	
7	No Resampling	–	DT	0.637	0.939	0.730	0.061	0.788	0.767	0.846	0.669	
8	Cluster-based only	1	SVM	0.591b	0.668	0.393	0.328b	0.629	0.579	0.620	0.388	
9	CluSMOTE	2	SVM	0.577	0.746	0.441	0.254	0.661b	0.60b	0.666	0.400	
10	CluSMOTE	3	SVM	0.498	0.790	0.486	0.210	0.644	0.580	0.675	0.396	
11	CluSMOTE	4	SVM	0.475	0.801	0.508	0.199	0.638	0.566	0.672	0.387	
12	CluSMOTE	5	SVM	0.468	0.819	0.529	0.178	0.643	0.572	0.683	0.401b	
13	SMOTE only	–	SVM	0.384	0.881b	0.606b	0.119	0.632	0.532	0.688	0.368	
14	No Resampling	–	SVM	0.409	0.874	0.569	0.126	0.641	0.557	0.699b	0.392	
Notes.

TPR True Positive Rate

TNR True Negaitive Rate

AUC Area Under ROC Curve

Gmean Geometric mean

a The best parameter value in DT model.

b The best parameter vaue in SVM model.

In CluSMOTE DT, AUC and Gmean have the same tendency. The best AUC and Gmean are 0.815 and 0.811 at r = 2 respectively. The AGm and F-measure values also have the same tendency, though the values are different. In DT, the best AGm and F-measure are obtained using the SMOTE oversampling method. In the SVM classifier, the best AGm is obtained using the SMOTE oversampling mechanism. However, the best F-measure is obtained using CluSMOTE at r = 5.

Previous studies on class imbalance stated that the hybrid resampling method could significantly improve performance. However, this was not the case in epitope prediction using the CluSMOTE DT method. No r value significantly influenced the overall performance improvement expressed by the AUC, Gmean, AGm, and F-measure. In case the TPR and TNR values are considered together, the selection of r = 2 is quite good as shown by the AUC and Gmean values. The selection of r values based on the experiment shows opposing conditions between the TPR and TNR. From Table 2, the best performance using AUC and Gmean is fairer compared to Agm and F-score. In the best AUC and AGm, a balanced proportion was obtained between the TPR and TNR. The best AGm and F-score resulted from the lowest TPR value. Generally, the performance models built with DT exhibit better performance than those from SVM. The performance of SVM is likely to be affected by kernel selection problems. Linear kernels are cannot separate the classes in polynomial cases. Other configurations or models may be explored for future work.

Comparison of the proposed method with previous methods

CluSMOTE DT was evaluated on an independent test set from Kringelum et al. (2012) by filtering the dataset from the details used in the training process of the method being compared. A total of 39 antigen data were used in the comparison, as listed in tab4. The final results of the test show that CluSMOTE with r = 2 is superior to the other methods with an average AUC value of 0.83. The average AUC values of Discotope, Ellipro, Epitopia, EPCES, PEPITO, and Discotope 2.0 were 0.727, 0.721, 0.673, 0.697, 0.746, and 0.744, respectively.

CluSMOTE DT with r = 2 was evaluated on the independent test set of glycoprotein antigen by Zhou et al. (2019). Testing with glycoprotein antigen showed that the performance of CluSMOTE DT was similar to that of SEPPA 3.0, with the AUC values of 0.766 and 0.739, respectively. Both CluSMOTE DT and SEPPA 3.0 were superior to Epitopia, Discotope 2.0, PEPITO, CBTOPE, SEPPA 2.0, and BepiPred 2.0. The detailed performance of the eight methods compared is shown in tab5. The AUC achieved by CluSMOTE DT is comparable to the one from SEPPA 3.0, showing that the proposed method might handle epitope cases with glycoprotein well. The model developed with CluSMOTE uses the dataset presented by Andersen, Nielsen & Lund (2006), which consists of 76 antigen structures. The number of complex structures used in the CluSMOTE model is less than that used in SEPPA 3.0, which consists of 767 antigen structures. The small number of antigen structures speeds up the training time for model development.

Conclusions

An epitope is a small part of the exposed antigen that creates class imbalance problems in the prediction of learning-based conformational epitopes. In this study, the CluSMOTE method was proposed to overcome the class imbalance problem in the prediction of the conformational epitope. The study shows that CluSMOTE considerably increases the TPR compared to SMOTE only. The comparison of the proposed model with state-of-the-art methods in the two datasets shows that CluSMOTE DT is comparable to or better than other methods. Its mean AUC values in Kringelum and the SEPPA 3.0 test sets are 0.83 and 0.766, respectively. This result shows that CluSMOTE DT is better than other methods in classifying the general protein antigen, though it is comparable to SEPPA 3.0 in the glycoprotein antigen.

Supplemental Information

Supplemental Information 1 Supplemental Tables

Click here for additional data file.

Supplemental Information 2 Exposed residue based on several Maximum ASA

Click here for additional data file.

The authors thank the Publishing and Publication Agency of Universitas Gadjah Mada for the English proof-reading of this manuscript.

Additional Information and Declarations

Competing Interests

Author Contributions

Data Availability

The authors declare there are no competing interests.

Binti Solihah conceived and designed the experiments, performed the experiments, analyzed the data, performed the computation work, prepared figures and/or tables, authored or reviewed drafts of the paper, and approved the final draft.

Azhari Azhari and Aina Musdholifah conceived and designed the experiments, authored or reviewed drafts of the paper, and approved the final draft.

The following information was supplied regarding data availability:

The data and source code are available at GitHub:

- https://github.com/BSolihah/LIBFROMJSAT

- https://github.com/BSolihah/conformational-epitope-predictor.

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
