# Peer review of "Enhancement of conformational B-cell epitope prediction using CluSMOTE"

_PeerJ Computer Science, doi:10.7717/peerj-cs.275_

## Round 0.1 · original submission · Major Revisions

Please check all the comments provided by the authors. I think it is excellent feedback to improve the paper. Also, make a thorough revision of English grammar (there are some typos and grammar mistakes in the text).

Reviewer 1 ·

Basic reporting

The English writing is poor.

Experimental design

no comment

Validity of the findings

no comment

Additional comments

I think this is a potentially interesting contribution on a challenging problem of solving the class imbalance problem in B-cell epitopes prediction. However, there are some major issues with the dataset construction, experimental design and English writing.
1.
The quality of the dataset need to be assessed. How is the sequence identity between any two proteins within the dataset? Any redundant sequences should be removed from the dataset to avoid bias.
As the author mentioned, the model validation was conducted by the 10-fold cross-validation method in the overall dataset. N-fold cross validation is not sufficient enough and another independent dataset should be used to test and compare with different sampling methods.
2.
I am confused by the purpose of this work.
If the author wants to prove ClustSMOTE to be an improved sampling approach, more datasets are needed to be used to compare with other sampling methods.
If the author would like to provide an method to solve the B-cell epitopes prediction problem, they need to compare their performance with other B-cell epitopes predictors.
3.
The text in this manuscript is often inaccurate or incomplete and needs to be revised.
In many places, the text is very sloppy/careless and often hard to make sense of, e.g., produced many features; Vaccine design experts use certain characteristics of residues to be identified; the risk of failure is still quite high; However, their performance level is still not high….
I suggest a careful rewrite for the whole paper.


4.
The method should be made publicly available in the form of a web-server or downloadable code. Either ways, the mode of availability should be indicated in the manuscript.


Minor:

1. Why the performance using DT is always better than SVM?
2. The full names of SMOTE, ASA and RSA should be explained.

·

Basic reporting

The article would benefit of a revision regarding writing style and phrasing. There are some ambiguities that make understanding what the authors intend to communicate challenging.

- Line 41, when you write "independent of expertise and precision", did you mean "dependent"?
- Line 45: "from others part of antigen" -> "from other parts of antigen"
- Lines 69-70. "the prediction of conformation epitope has not been explored..." and "feature extraction methods..." How are both facts related? If the same data is used for both, why the lack of a dataset affects only to the prediction and not to feature extraction/balance? I would suggest rephrasing lines 69-72.
- Line 83: I suggest to briefly introduce the basics of each category.
- Line 87: clustering-based approaches belong to the data-level category, correct? I suggest to specify this in the text.
- Line 89: "approaches used which are regardless" -> "approaches used: regardless"
- Line 125: "were" -> "where"
- Line 127-128: "item data" -> "samples"
- Line 129: "surrounding" -> "resulting"
- Line 129: If one class contains 944 samples and the other contains 12464, why is the ratio imbalance 1:10? Wouldn't it be 1:12/1:13?
- Lines 137-138, I suggest to rephrase so that the explanation is easier to follow
- Line 143: "used feature" -> "applied the feature"
- Line 158: shouldn't the names of the features be in bold too? Also, why is Contact Number (CN) not in Table 2?
- Line 184: "However, it also has an advantage", please clarify which advantage?
- Lines 184-185: "advantage", perhaps you mean "disadvantage"? And which is the advantage of undersampling?
- Line 255, "DT": this is not the first time in the document that you refer to the decision tree, the acronym should be defined earlier.
- Line 282: I suggest to present the name that you give to your method (CluSMOTE) earlier, and to use through all the paper. As you are comparing your proposal with regular SMOTE and clustering, it is at some points difficult to follow which method are you referring to.
- Line 336: "Data is available at", there seems to be some text missing here.

I suggest to add some more references, specially:

- On lines 55-61 you provide several references to characteristic-based approaches, which you seem to be focused on. Then, on lines 62-65 you barely give any examples on graph-based (only two cites, and both from the same author). Is this because there are not many works falling in this classification. If this is the case, you may considering stating so. If not, consider adding more references.
- On lines 65-68, please provide some references on learning-based approaches.
- Line 80, please provide references for each method.
- Line 85, please provide a reference for random undersampling.
- Lines 153-154, please provide the reference where it is concluded that there is no difference.
- Lines 165-166, I suggest to put the URL as footnote, or to cite it using url citation style instead of including it on the text. It is also relevant to include the date when you accessed that URL.
- Line 189: please provide reference for HDBSCAN.
- Line 279: please provide reference for k-medoid.

Experimental design

The explanation of the methods should be extended to provide enough information to make it repeatable.

- Lines 193-194, further explanation is needed: how are you defining "optimal" here? what metrics are you using to guide this step?
- Line 197: consider using the equation editor (if using Microsoft Office) or the adequate latex library for writing your formulation. Also, explanation on several parameters is missing (you only define r).
- The parameters used for the classifiers (SVM and DT) need to be detailed for repeatability. Moreover, several approaches have been proposed for both methods through the years, so I would suggest including references for exactly which configuration of SVM and DT are you using for your experiments.
- Section starting in line 205. AUC and TP, TN, FP, FN definitions are widely known, and I don't think is necessary to include the equations 1-3
- In equation 6, there are some parameters that are not defined.
- Line 237: does this mean that you are computing the average of the validation set for the 5 times that the process is carried out? please clarify a bit these sentences.
- Lines 239-241, please state the software versions.
- Line 241, please detail the modifications, for repeatability purposes.
- Line 248, you state that "r=1 states that the number of the negative class dataset is the same as the positive class or without SMOTE". I understand that r=1 means that only the clustering, undersampling step, is applied, correct? I suggest to rephrase it to make it more clear.

In general, I would suggest that, when referring to results that are in a Table, you should explicitly reference the table.

- Lines 260-261, "did not affect the performance of the algorithm", however in Table 4 the values are slightly different.
- Lines 265-268, further clarification is needed. What are the optimal conditions in this context?

- Table 2: I suggest to highlight the best value for each column in bold.
- Table 3 is missing the units (I assume that they are milliseconds, as in the figures, but you should indicate it). I would suggest to remove this table entirely, because it does not add much if compared to the figures (and it is easier to observe the trends in the figures).
- Table 4: why only the columns for AUC, Gmean, Adjusted Gmean, and Fmeasure are highlighted? I would highlight the best values in all the parameters/columns. Also, it would be better to indicate the r value in each case.

In the experiments, I would suggest to include also the values without applying balancing methods (that is, the results from trying to classify the original dataset as-is), so that we can observe the improvement achieved by the proposed method over the original.

Validity of the findings

- Line 257, you mention that "the increase in r value decreases TPR and increases TNR", but this only applies to DT, not SVM. I suggest to clarify this in the text, and also to explain why the classifiers behave differently.
- Lines 258-259: "this is in contrast with the use of the SMOTE method". However, your results varying r (Table 2) do not compare regular SMOTE and your approach. The only results that actually compare both approaches are in Table 4, and you use only 1 r value. You would need to include more results or references to support that regular SMOTE behaves differently than your approach.
- Line 260: "from the AUC parameter, the optimal r value = 2", this also holds true only for DT, while in SVM higher r values seem to yield better results.

- Figure 2: I find these results a bit odd. I would need to know the specifications of the algorithms (model, parameters, if the software and hardware were the same in both cases), but I would recommend the authors to double-check the training of the SVM and, if the values are indeed correct, to provide an explanation of why is this happening. In any case, you need to add the specifications of your computer for this data (Figures 2 and 3) to be meaningful, as now it is unclear how to interpret your times. Moreover, perhaps it is better to change the scale of Figure 2 to seconds instead of milliseconds?
- Figure 3: do you have any explanation for the behavior of the SVM? it seems to get faster when the ratio increases, but from 3 to 4 it became slower again.

Additional comments

The main points which I think would benefit this article the most are:
- Review of the phrasing and style, to improve clarity. For example, for consistency, define the terms the first time they appear (specially for method's names) and then use them through the whole document
- Extend the methods: more information is needed, the article is missing information for it to be fully repeatable

Reviewer 3 ·

Basic reporting

Cluster-based undersampling and SMOTE for class imbalance on conformational B-cell epitopes prediction method

Binti Solihah Corresp, Azhari Azhari, Aina Musdholifah


Handling class imbalance is a chalanging problem in conformational epitope prediction. This study proposes a cluster-based sampling method to overcome the chalange. To that end, Machine Learning methods (SVM, RFC) have been applied for the classification purposes with the implementation of SMOTE and the performance is compared (by means of a 10-fold cross-validation) to the SMOTE, Random Undersampling, and cluster-based undersampling. The results shows improvement over earlier methods (AUC and Gmean parameter is 0.87% while Adjusted Gmean achieved 0.88% score and F-measure achieved 0.53% score). My overall assessment is that its an okay paper which requires major revision to meet the publication standard of the journal and the corresponding field.

The first lecuna of the paper I find is that it is full of abbriviations! The authors should explicitely spell out several of such abbriviations when they are first mentioned! The other shortcoming is that they don’t take care to explain a few basics in the intro for example what is «class imbalance»? What is «random undersampling»? Where is the cross-reference for such terms? Do the authors consider their readership of their paper to be restricted to a few real exparts in the field alone?

1. What SMOTE stands for ? Should be declared in the abstract. What is the underlysing principle / philosophy of this methodology – should be described in the intro.

2. CEP: utilizing the characteristics of solvent accessibility (Kulkarni-Kale et al., 2005) (Introduction, Line 56) – should be rephrased as ‘solvent accessibility based properties’.

3. Discotope: combining characteristics of solvent accessibility and log-odd ratio – should likewise be ‘ solvent accessibility based properties and log-odd ratio’! ‘log-odd ratio’ of what?

4. what is AAIndex? Amino Acid Index?

5. ‘However, it is constrained by many problems such as imbalance classes that occur in the formation of datasets for the learning process.’ - ‘imbalance classes’ should be ‘imbalanced classes’!

6. imbalance class handling should be imbalanced class handling or handling of class imbalance!

7. The defination of ‘negative (non-epitope) and positive class’ in the given context is given much later before the terms are used for the first time! (Line 93)

9. Overall, the introduction requires substantial rewriting to make it more lucid and readable. It is too full of gergonswithout proper defination / references !

10. Features Extraction: ASA: The state-of-the-art to compute ASA is the Lee and Richards method implemented in NACCESS. The authors should explain the choice of their PSAIA software (Mihel et al, 2008).

11. RSA is also (better) known as burial of solvent accessibility (Basu et al., BMC Bioinformatics, 2011, 12:195). The authors should definitely mention that they are the same.

12. Calpha should be Cα - wherever applicable.

13. Protrusion Index: The percentage of residual atoms in an ellipsoid centered on atoms exposed
in the outermost position (Ponomarenko et al, 2008). - the sentence is incomplete. How residual atoms are defined in this context?

14. B-factor states atomic mobility – due to thermal vibrations!

15. The equations should be better formatted using proper equation editor.

16. «In order to test the reliability of the resulting model, the model validation was conducted by the 10-fold cross-validation method in the overall dataset. Each cross-validation process was carried
out five times, for every process the corresponding performance is recorded and expressed in the mean and standard deviation.» - why five times? To check stabilityof the results? Then 5 is a very low number. The authors should explicitely mnention the random components that might differ between each run effecting the change in the overall performance. Usually an n-fold cross validation is good enough and can be complemented by another independent validation (see: https://www.ncbi.nlm.nih.gov/pubmed/27307625). To my view, the authors should carry out an independent validation and compare the performance with the competing methods – if an indendent benchmark is available.

17. After all this effort, what is the net percentage increase in performance (say, based on AUC) of the current method with respect to the privious state-of-the-art? Should be explicitely mentioned.

Experimental design

Some of them should be revised as suggested in the 'basic reporting'

Validity of the findings

suggested in the 'basic reporting'

Additional comments

The authors should carefully consider addressing all points raised in the review.

Annotated reviews are not available for download in order to protect the identity of reviewers who chose to remain anonymous.

---

## Round 0.2 · Minor Revisions

Please follow the recommendation of both reviewers, including a report showing how these recommendations were addressed.

·

Basic reporting

The goal of the manuscript is to tackle one of the main drawbacks in the current epitope prediction methods: class imbalance. To that end, they propose a method that combines undersampling of the majority class using clustering, and oversampling of the minority class applying SMOTE. The authors compare their outcome with other methods for epitope prediction. They also compare the results obtained applying two machine learning classifiers to the data resulting a) applying only the undersampling part of the method b) applying only the oversampling part of the method and c) applying the complete method, with different configurations.

The quality of the manuscript has been much improved since the previous review, and there are only some minor suggestions that I would like to share with the authors.

- Is HDBScan an acronym? if so, explain at first occurrence.
- Same for SVM and DT, use the complete name once before starting to use the abbreviation.
- Line 60, you refer to the last group as "machine learning", and then in Line 78 as "learning-based". I would suggest to choose one of them, and use it always in the other references to the same group.
- Line 89, class imbalance could be defined in a more general way as when the amount of samples from one of the classes is larger than in the other (if the ratio positive/negative classes were 10:1 instead of 1:10, that would still be an imbalanced dataset, and lead to classification issues).
- Lines 103-104, you can remove the sentence "Many methods have been proposed to overcome the class imbalance problem", as it is a repetition from Line 93.
- Line 114, "Providing different weights is categorized as cost-sensitive in the algorithm level". Is this still within the algorithm-level group or to introduce the cost-sensitive group? If it is the second, I would suggest to state so, as you did with the previous groups ("In the algorithm-level approach...").
- Line 120, "top", change to some more specific, such as "most common" or "best performing", depending on what you meant.
- Line 212, "An exposed residue is likely to bind to other molecules for accessibility reasons", this sentence does not add much information on its own, and does not seem to be related with the explanation that follows.
- Line 218, "a number of" -> "several"
- Line 252, if the file is available as supplementary material, please state so.
- Line 319, could you provide a reference for the decision stump learner?
- Line 326, could you add a reference for the C4.5 algorithm?
- Line 342, I would suggest to translate "jika" to either English ("if") or mathematical notation
- Line 352, is the word "complex" equivalent to "fold" in this context?
- Line 431, "better than other methods in the general protein antigen" -> "better than other methods in classifying the general protein antigen"

Experimental design

- From the abstract, it is still a bit unclear what classification task are you performing, I would suggest to state it more clearly.
- Line 206, please clarify why do you choose that specific value.
- Line 258, "sliding window 9" -> "sliding window of size 9". Was this value taken from a previous work, or determined empirically for this experiment. Please clarify in the text.
- Lines 323-326, I would suggest to not refer to the specific functions within the software that you are using, and substitute that for the more general algorithm concepts (e.g. do not specify Integer.MAX_VALUE, just state that the maximum depth of the tree is 10). This makes your method repeatable for readers that use different software.
- Lines 401-420: specify what r value is used for these experiments.

Validity of the findings

- Line 372, "This tendency shows that the more negative class data that are sampled, the lesser the loss of information". Do you mean that with larger r you undersample less, so you remove less information from the data/the model has more information to work with? I would suggest to rephrase this sentence and clarify it.
- Line 395, are you referring to both combined? otherwise, TPR decreases, and TNR increases in a similar proportion. The AUC and the Gmean are the best for r = 2, and the FPR also improves at r = 2 if compared with r = 1.
- Lines 398, "performance of SVM is affected by kernel selection problems" -> if this is an hypothesis from your results, rephrase to "performance of SVM is likely to be affected by kernel selection problems" or "in view of the results, the performance of SVM is probably affected by kernel selection problems". I would also suggest to include a remark in your conclusions stating that other models or configurations may be explored, and that your classification results may become even better, as the classifier fine-tuning was not the main goal of this work.
- Line 427, "considerable increase in TPR", please clarify what are you comparing to: previous works, using regular SMOTE...
- Table 2: there is a typo on "Precision". In the column PPV, why is the 8th row highlighted? Also, in the column FPR, you want the false positive rate (FPR) to be as small as possible, correct? the highest value for each model is now highlighted in that column.

Reviewer 3 ·

Basic reporting

Although the authors have revised the manuscript there still are issues that need to be addressed.

1. "The disadvantage of NACCESS is that it can only count one molecular structure at a time; thus, users must create additional scripts to count a number of molecular structures at a time (Mihel et al., 2008)." (Line 216)

"The PSAIA is not only limited to counting one molecular structure but can also be used to calculate other features, such as RSA, PI, and DI." (Line 219)

- So the authors mean to say both NACCESS and PSAIA have the same limitation that they can only count one molecular structure at a time. Then, why are they using PSAIA rather than the more standad NACCESS method - is not clear at all, especially given the fact that they themselves elaborate on the different ASA values returned by NACCESS.


2. The use of the Gly-X-Gly peptide fragment is relevant to compute burial of solvent accessibility (bur) - which is required if you want to demarcate individual residues into burial bins (buried / exposed). I am not sure as to whether they require residue-wise burial for a similar purpose or that the purpose can be served by ASA for the residue surface patch alone without normalizing it in a residue-wise manner! So, I am not sure whether the GXG tripeptide theory works here! It's not very clear from the statement: "whether a residue is likely to bind to other molecules for accessibility reasons"! In that case, what are the criteria they have adopted for declaring a residue completely buried or partially buried or completely exposed - must be elaborated!

I would deeply encourage them to use the already set standards in the field as in

Basu et al., BMC Bioinformatics, 2011, 12:195

0.0 <= bur <= 0.05 : Completely buried (bin1);
0.05 <= bur <= 0.15 : Partially buried with higher burial (bin2);
0.15 <= bur <= 0.30 : Partially buried with lower burial (bin3);
0.30 <= bur : Completely exposed (bin4);

Experimental design

N/A

Validity of the findings

The suggested minor revision is a must. Then only it can be validated.

---

## Round 0.3 · Minor Revisions

Please follow the last recommendation to prepare the new (and hopefully definitive) version

Reviewer 3 ·

Basic reporting

I am extremely displeased and taken to a surprise that the authors don't really care to cite the source article (Basu et al., BMC Bioinformatics, 2011, 12:195) proposing the standardization of RSA bins even after acknowledging the same in the rebuttal.

The original point raised in the review:
* * *
I would deeply encourage them to use the already set standards in the field as in

Basu et al., BMC Bioinformatics, 2011, 12:195

0.0 <= bur <= 0.05 : Completely buried (bin1);
0.05 <= bur <= 0.15 : Partially buried with higher burial (bin2);
0.15 <= bur <= 0.30 : Partially buried with lower burial (bin3);
0.30 <= bur : Completely exposed (bin4);
* * *
Author's reply:

In the study of conformational epitope prediction, the RSA value is used for two purposes: (1) creating a dataset to form a classification model. (2) as a feature in the formation of classification models. Related to the formation of the dataset, Rubinstein et al. (2009) use the definition of RSA> 0.05 to get the residue exposed (using the standard as used in Basu et al. (2011)). In Kringelum et al., (2012), the RSA limit was 0.01,0.05 and 0.5. Ren 2014 uses RSA> 0.25 because 75% of epitopes have RSA> 25.9%
* * *
The burial bins were first formalized by Basu et al., BMC Bioinformatics, 2011, 12:195 and therefore it is imperative that any resetting of the bins afterword (in other words, if the methodology is used in a variant form) should cite the source article. This is very very common in academic publishing and a violation of this, to my modest opinion qualifies as unethical.

I would sincerely request the editor to strictly look into this matter.


The rest of their responses are okay with me, though there are several grammatical flaws that should be corrected.

Experimental design

Same as 'Basic reporting'

Validity of the findings

Trustworthy and logical

Additional comments

Please refer to the basic reporting.

---

## Round 0.4 · accepted · Accept

This version of the paper addresses the concerns of the reviewers. Now, the paper is ready for publication. Congratulations